# Liquidity and Corporate Governance

**Tom Berglund**

Department of Accounting and Corporate Law, Hanken School of Economics, Arkadiagatan 22, 00100 Helsinki, Finland; berglund@hanken.fi

**Abstract:** This paper discusses the relationship between stock market liquidity and corporate governance. Both concepts are widely investigated from different angles in the literature. It is generally agreed that they are related so that better corporate governance implies higher liquidity for shares of listed companies. However, the importance of good corporate governance for the market liquidity of the share will differ depending on the characteristics of the firm's business. Good corporate governance will be particularly important in reducing agency problems in firms where the business is subject to a high degree of uncertainty. Proper corporate governance, in other words, matters most for firms where external assessment of the firm's business prospects is difficult, while it is less important for value creation in firms where the business is easier to understand.

**Keywords:** asymmetric information; board composition; information disclosure; market microstructure; price informativeness

---

## 1. Liquidity and Corporate Governance

Liquidity is generally seen as a desirable property for the shares of a listed firm. Better liquidity makes it easier for the owner to sell some of the shares, should an unforeseen need for cash arise. Better liquidity also makes observed prices more reliable since more competition between traders on both sides of the market will cause any temporary mispricing to vanish more quickly. As for corporate governance, good corporate governance is also a desirable feature of listed firms. Good corporate governance implies that the shareholders' interests are reliably taken into account. Good corporate governance will accordingly reduce the share price discount caused by investors' perception of how likely it is that those who run the firm will extract money from it at the shareholders' expense in some more or less sophisticated way.

This paper focuses on to what extent high liquidity and good corporate governance, both being in shareholders' interests, do reflect the same fundamental firm characteristics. The value relevance for shareholders of both properties stems from risk caused by information asymmetry as perceived by outside investors. Better corporate governance will reduce information asymmetry and reduced information asymmetry risks will increase liquidity. Good corporate governance will discourage management from exploiting its information advantage to the detriment of present or future shareholders. Having good corporate governance in place will thus be more essential in firms with a broader scope for moral hazard on behalf of management than in firms where managerial misbehavior is relatively easy to detect by an outside investor.

In the following, the concept of liquidity will be discussed, first in order to arrive at a definition that seems consistent with how the concept has been commonly used in the literature. Next, what "Good Corporate Governance" as a concept stands for will be discussed, with the same goal in mind. The role of corporate governance in reducing costly information asymmetry for outside investors is then discussed. Firms operating in a rapidly changing business environment are argued to be more dependent on good governance than firms in stable environments. The challenges to corporate governance in trying to

properly address information asymmetry that is detrimental to firm value will finally be considered. A brief summary concludes the paper.

## 2. Liquidity

Liquidity is an important aspect of any market for financial securities. With the "liquidity" of an asset in general, we understand how easy it is to buy or sell the asset at any point in time. The less costly it is to transform the asset into money, or vice versa, the more liquid is this market. A closer look reveals that there are several relevant dimensions to liquidity. Following Kyle (1985), there is what he calls "tightness" i.e., how much it costs to turn around a position in the asset over a short period of time, "depth" i.e., how large an order flow innovation has to be to change prices by a given amount, and "resiliency", which is how rapidly prices recover from a random, uninformative shock. Even if these three aspects are correlated, they are not identical.

A well-established insight from actual markets is that liquidity tends to be related to presumed information asymmetry concerning the asset. Since Akerlof (1970) seminal paper, it is well known that information asymmetry will give rise to expected trading costs for uninformed market participants. The reason is that a higher degree of information asymmetry for an uninformed market participant implies a higher likelihood to end up trading with an informed counterpart who happens to know that the present transaction price differs from the actual value to the counterpart's advantage. This information asymmetry, as Akerlof (1970) explained, may result in a total breakdown of the market for the good. Market failure can normally be avoided, but the information asymmetry must be taken into account by a potential uninformed market participant as part of expected trading costs in that market.

Many formal models that incorporate these expected information asymmetry costs have been presented; the model by Kyle (1985) being among the first. Easley and O'Hara (2004) show that in an incomplete asset market with private and public information, an equilibrium exists in which shares with more private information command a higher expected return than shares with less public information, due to the difference in the risk for an uniformed investor to end up making a trade with a better informed counterpart. In a companion empirical paper, Easley et al. (2002) show that their measure of the probability of conducting a trade with an informed counterpart, the so called PIN measure, first proposed by Easley et al. (1996), indeed provides a significant explanation for differences in future stock returns in large cross sections of stock returns for US firms. The time varying nature of prevailing information asymmetry and how it is captured in a dynamic version of the original PIN measure is discussed in Easley et al. (2008). Easley et al. (2010) report results in support of a significant priced dynamic PIN-factor on US stock returns from 1982–2002. For an example of independent results in support of a priced PIN measure see Agudelo et al. (2015), who found that the dynamic version of the PIN measure seems to capture information asymmetry in six different Latin American markets. However, in empirical studies of the market liquidity of an asset, it is important to keep in mind that all actual measures of the degree of information asymmetry for a given company's shares at a given point in time are likely to contain considerable measurement problems.

## 3. Corporate Governance

In the following it will be taken for granted that the corporate board is responsible for the quality of corporate governance of the firm. The board is assumed to be there to promote value creation as the goal for the firm's activities, or more precisely, to maximize the long-run value of the equity in the firm. As argued among others by Michael Jensen (2001), if the firm is run in the long-run interests of its shareholders, it will normally also act in a way that is consistent with its other stakeholders' interests. Any attempts by a firm, that faces competition in its output, as well as input markets, to exploit customers, employees, or other input providers will make these counterparts shun the firm, and a loss for the firm and its shareholders will consequently occur.

Promoting value creation by the firm is far from a trivial task, though. The reason for this is vividly described in the following quote from the Economist:



> *AIRPORT BOOKSHOPS teem with guides that promise to teach executives the secrets of success. Read this tome, follow this philosophy, change your habits and you too can be a management titan. As a moment's reflection on business history demonstrates, there is no sure-fire route to glory. Instead, running a company is a permanent exercise in juggling trade-offs. What is the right course of action may vary at different times, and in different industries.* The Economist (2019)

This description of the job that executives are doing highlights why the task of the corporate board is so difficult. The board must regularly judge whether the CEO of the firm and the top management team is doing a proper job in "juggling" the relevant "trade-offs" in the firm's day-to-day business.

Shleifer and Vishny (1997), in a widely cited survey article, claim that the main role of corporate governance is to reduce moral hazard on behalf of management, in effect to reduce the likelihood of attempts by managers to enrich themselves at the expense of shareholders. This puts the focus on possible attempts of top managers to enrich themselves at shareholders' expense. In the wake of the much-publicized corporate scandals, like Enron and WorldCom in the USA, corporate governance came to a large extent to be seen in a more limited sense as a mechanism by which fraudulent activity by the top management can be discouraged. Increasing the likelihood of early detection of any fraudulent activity will naturally reduce the temptation for the top management to cheat the firm's shareholders.

However, losses of a much larger magnitude than those caused by outright fraud[1] may accrue to the shareholders of a firm if the top management, for personal reasons, bypasses large projects with huge potential impact on the firm's future profits. Such projects could be radical restructuring in response to introduction of new technology, acquisition of extensive complementary production capacity, and spinning off major parts of the present operations of the firm at a significant premium. The expected value increase to shareholders from these types of projects may simply fail to adequately compensate for the required additional personal effort from management and for exposure to risk of a possible project failure. Since the top management has a superior overview of the firm's business environment, it can deliberately play down the likely benefits and exaggerate the risks of the project and thus justify the decision not to embark on the project. For nonexecutives on the board, who are not intimately involved in charting strategic alternatives for the firm, such biases may be hard to detect, and even more difficult to convincingly pinpoint in a board meeting.[2]

Of key importance for board efficiency is to safeguard that as much relevant information as possible will reach the board. As pointed out by Adams and Ferreira (2007), the top management is in a position to withhold crucial strategic information from the board. As a result, an overly critical attitude by the board towards the top management will most likely make management less willing to reveal any unfavorable information. An efficient board must pay close attention to how the management can be induced to update the board also on issues that may result from past management mistakes, and about strategic alternatives that management for personal reasons may dislike.

In summary, proper handling of corporate governance issues in today's world is a demanding task. Board members that understand the essential features of the business at hand, and are good at expressing their views, are needed to properly address that task. In addition, these persons have to possess exceptional personal integrity to have the courage to bring up observations that are likely to be unpleasant for the top management of the firm.

---

[1]  Shleifer and Vishny (1997) recognise this too, e.g., they write: " … perhaps most important, managers can expropriate shareholders by entrenching themselves and staying on the job even if they are no longer competent or qualified to run the firm."

[2]  Avoidance of expected value increasing projects is a particularly relevant form of executive moral hazard in today's world, when rapid technological development creates new opportunities and renders old production solutions obsolete at an ever-increasing pace. In such an environment, proper judgement on how willing and capable the management is to handle relevant challenges imposed by the rapidly changing business conditions require much more than careful reading of financial reports and other information that the firm is required to disclose. To complicate matters, as argued in Berglund (2015), mandating management to publicly disclose all information needed to make proper judgement on the management's abilities will not always be in the best interest of the firm's shareholders. For instance, pieces of information that would clearly benefit competitors should, in the best interest of the firm's owners, usually not be disclosed.

## 4. How Liquidity Relates to Corporate Governance

The relationship between the quality of the corporate governance of a firm and the market liquidity of its shares has been subject of many well-known studies in finance. In an influential early article, Coffee (1991) argues that liquid stock market trading of a firm's shares will discourage institutional block holders from engaging in the corporate governance of the firm. The reason, according to Coffee (1991) is that with a liquid market it is cheaper for the block holder to sell its shares than to actively orchestrate a change in cases where the block holder observes corporate governance issues that need to be addressed in that firm. However, this view is challenged by Maug (1998) who notes that with a liquid market a block holder can reap larger benefits from addressing corporate governance issues in a target firm simply by increasing its holdings without a mitigating impact on the share price. Hence, Maug (1998) concludes that there is no conflict between market liquidity and vigilant large shareholders that monitor the firm's management. In support of this conclusion, Norli et al. (2015) show that higher liquidity of a firm's shares significantly increases the likelihood that the firm will be subject to shareholder activism in a sample of US firms during 1994–2007. Helling et al. (2019) show that the impact of large institutional owners on R&D investments is higher in US firms with more liquid shares. John et al. (2019), analyzing the consequences of a reform that substantially increased the liquidity of some shares in the Chinese stock market, conclude that large shareholders responded by improving corporate governance of their firms rather than selling off their shares.

Furthermore, Holmström and Tirole (1993) point out that better liquidity allows the board to use the share price to more efficiently reward the top management for decisions that increase firm value. Better liquidity, in other words, allows more precise alignment of interests between top management and shareholders. Consistent with this observation, Feng and Yan (2019), on US data covering 1992–2015, found that CEOs with high pay-for-performance sensitivity in their compensation contract exert extra effort in improving the firm's information environment so as to improve the liquidity of the firm's shares.

The underlying factor that connects the market liquidity of a listed firm to its corporate governance is information asymmetry. If uninformed investors believe that likelihood of ending up in a trade with a better-informed counterpart is high, they will require an expected compensation for this risk as observed, among others, by Easley and O'Hara (2004). Thus, a share that is subject to less information asymmetry in the stock market will trade at a higher price than a similar share subject to more information asymmetry. Good corporate governance accordingly requires that the corporate board favor timely and accurate information disclosure. Firms with good corporate governance will be subject to less information asymmetry than firms with bad corporate governance. Furthermore, since less information asymmetry will reduce expected trading costs from adverse selection, trading in the shares will also be more attractive, resulting in higher liquidity.

The hypothesis of a positive relationship between corporate governance and liquidity is supported by a large number of empirical studies. Chung et al. (2010), using a broad sample of New York Stock Exchange and Nasdaq listed firms in the USA, show that there is a clearly significant relationship between a broad-based measure of the quality of internal corporate governance, a measure that they construct based on data from Institutional Shareholder Services, and the liquidity of trading the firm's shares. This relationship is remarkably robust when controlling for confounding factors.

Lee et al. (2016) show that the same result holds when analyzing the informational efficiency of the stock price instead of liquidity. Better corporate governance implies more accurate stock price reactions to new information. This further supports the conjecture that good governance puts stronger pressure on management to release accurate and timely information.

Chung et al. (2012), based on data for 25 countries from 2003 to 2011, found that common law countries tend to have better corporate governance and higher stock market liquidity than civil law countries. They also found that shares in firms with better corporate governance tend to be more liquid than shares in firms with worse corporate governance, regardless of the legal origins in which the firm operates. On a large sample of Australian firms in 2001–2013, Ali et al. (2017) found a

significantly positive relationship between corporate governance quality of firms and their stock liquidity. They also found that corporate governance quality improves liquidity since it is associated with higher information disclosure.

For the UK, Lehmann (2019) reports results from a natural experiment on this issue in a carefully done study of a "joint indices project" between ISS and the Financial Times Stock Exchange (FTSE). This project increased the ISS corporate governance coverage, going from 2004 to 2005 in the UK, from 212 firms up to 524. He found that the coverage initiation led to "improvements in governance quality, liquidity, financial analyst following, and investor breadth". These results were stronger for firms that before the initiation had weak governance, liquidity, and financial analyst following.

For a positive relationship between corporate governance quality and liquidity, it is essential that those in charge of the corporate governance of the firm perceive their role as being equally responsible to shareholders who are buying the firm's shares, as to those present shareholders who are selling[3]. Any attempts to manipulate the share price either up or down will hence be unacceptable. From a corporate governance point of view, the firm's information dissemination should give an unbiased picture of the future prospects of the firm. At disclosure of new information regarding a significant event, the focus should be on its likely impact on expected cash flows of the firm, combined with an accurate assessment of any risk exposure impact of the event.

Strict application of this, what could be called *unbiasedness principle,* will be beneficial for the outside investor that would like to invest in the firm's shares (who lacks any access to classified information). The likelihood of paying too much to an insider who knows that the firm happens to be overvalued will be relatively low. As a consequence, the interest among outside investors to buy such shares will be higher than for firms where adherence to this unbiasedness principle is in doubt.

A board that efficiently pursues this principle will carefully monitor that no one within the firm, in possession of relevant news at an early stage, will make any attempts to exploit temporary information advantages. Good corporate governance can, in this respect, be taken to promote the same objective for the firm as insider trading regulation does for the market as whole. Trading on privileged information is strictly forbidden in stock markets for the reason that we just discussed. A strict ban on trading based on inside information will reduce the likelihood that an outside investor ends up trading with someone who happens to know that there is a temporary pricing error due to new information that has not yet been incorporated in the price. Insider trading regulation addresses the market as whole, while corporate governance is firm specific. While insider trading regulation targets misuse of specific value relevant information, corporate governance is concerned with dissemination of information to outside investors about the firm and its prospects in general.

In summary, the impact of corporate governance on the liquidity of the firm's shares in the stock market comes from expected reduction of any biases in the information that the firm disseminates to the market. If investors believe that there is a genuine will among those in charge, i.e., the board, to uphold an unbiased picture of the future prospects of the firm, that is to avoid a skewed assessment in one direction or the other, the information asymmetry costs will be relatively small. Consequently, trading in the firm's shares will be more attractive than if investors suspect that deliberate attempts by management to distort the prospects will be quietly accepted by the board.

## 5. Unbiased Information Dissemination as a Challenge for Corporate Governance

The most important motives for promoting a biased view of the firm's prospects are likely to be personal motives of the executive management. Even in the absence of share-price-related compensation schemes, executive management may have incentives to promote a biased view of the firm's prospects. The willingness to provide loans to the firm, and in general to conduct business with the firm, is partly dependent on how the future prospects of the firm are perceived by the counterpart.

---

[3] See Michael Jensen (2001).

As a result, there is a natural tendency for the firm's executive management to "sugar coat" news so as to create an overly favorable impression here and now of the future prospects of the firm.

With share-price-related performance compensation schemes, the temptation for management to deliver an overly favorable view of the firm's future business will be strengthened further. From the board's perspective, this implies that higher intensity of the share-price-related incentives for the top management must be combined with more attention to potentially misleading, overly optimistic, information dissemination.[4]

One may argue that subtle biases in the information that the firm provides for investors is difficult to avoid, and that attempts to spot such deliberate attempts in advance are likely to be challenging. However, this makes it even more important for the corporate board to handle the challenge appropriately. The main incentive for the board members, in their turn, to tackle this challenge properly consists of potential loss of reputation. In case of a clear failure of the board to prevent disclosure of biased information, which later happens to get exposed by media, the personal costs for the board members could be substantial. The tarnished reputation will reduce the board member's future opportunities and expected future personal income.

As the seminal article by Hermalin and Weisbach (1998) clearly explains, the role of corporate governance and the board will not be the same in all types of firms. When it is easy for outside investors and financial analysts to estimate the future cash flows from the firm's business, like e.g., in a real estate investment trust, the role of the board in safeguarding unbiasedness will not be that important. In such cases, the opportunities for management to promote a skewed picture of the firm's future business prospects will be strictly limited. Any attempts to distort the market's perception would lead to a loss of credibility for that management in the stock market, since such attempts most likely would be detected by independent financial analysts.

In contrast, when there is substantial uncertainty concerning the future of the firm's business, e.g., due to an ongoing industry-wide introduction of a disruptive technology, reliable detailed estimates of future profits for individual firms will be difficult to obtain. In such situations, the temptation for the management to provide overly optimistic assessments of the future prospects of the firm could be substantial. For investors that consider going long in such a firm's shares, the presence of a board with alert members, that have their personal reputation at stake, may be what tips the scale in favor of investing.

Proper institutions will also matter in corporate governance. Since talented managers must be skilled in convincing people around them of what to do, it is important that there are strictly enforced rules that constrain their use of this skill. Otherwise, managers that initially are quite successful may at some point start to entrench themselves, preventing required changes in response to technology-driven shifts in the business environment.

Around the world, institutional rules that embody what has proved beneficial in this respect, are spelled out in official corporate governance recommendations for listed companies[5]. In spite of considerable differences in how these recommendations are structured and formulated, their main proposals are largely similar[6]. One such recommendation is for the presence of independent members on the board. Having members on the board who are independent of management is essential

---

[4] The temptation to manipulate the share price in order to increase the value of executive call options is, of course, well known. Perhaps the best-known example is the Enron case (see e.g., Slater 2008). In practice, the main way in which this potential problem has been addressed is through requiring a long enough vesting period before the executive stock options can be exercised, the idea being that strategies to boost the stock price in the short run are relatively easy to come up with, but most of them will generate a reversal later on when evidence that reality is less favorable accumulates.

[5] The OECD principles of corporate governance (OECD The Organisation for Economic Co-operation and Development), are a widely used reference that have influenced local recommendations around the world.

[6] Empirical studies concerning the impact of individual recommendations are notoriously difficult to conduct. Measurable characteristics are few and the measures quite crude. Furthermore, most studies suffer from the so-called endogeneity problem, stemming from the fact that incentives to comply with existing recommendations are likely to vary depending on the performance of the firm (see Bhagat and Black 2002).

in trying to safeguard that the management avoids decisions that would benefit themselves at the shareholders' expense.

As shown in the empirical studies surveyed by Yermack (2006), the stock market pays attention to appointment of board members. Yermack (2006) concludes that "share price reactions are sensitive to variables such as a director's occupation, independence, and professional qualifications". This finding points to the fact that strict compliance with all corporate governance recommendations will not guarantee good corporate governance. For instance, formally independent board members can differ a lot in terms of how efficiently they are doing their job on the board, something that the market reaction to appointments roughly anticipates.

Ferreira et al. (2011), using a highly simplified model in the spirit of Hermalin and Weisbach (1998), show that the relationship between board independence and price informativeness is likely to be ambiguous. The reason according to them, in essence, is that board monitoring is an expensive activity. To do a good job, the persons involved have to spend substantial effort on the task. They also have to possess scarce talent that is likely to carry a high alternative cost in the market. Thus, in equilibrium, we would not expect the most skilled and thus expensive talent to be engaged in firms where the marginal contribution of this talent is relatively low, as in firms where well-functioning other mechanisms, like analyst-following and an active market for corporate control, efficiently discourage managerial moral hazard. Based on their empirical analysis, Ferreira et al. (2011) conclude that their " . . . results suggest that firms with more informative stock prices have less demanding board structures".

Earlier work in the same spirit includes Raheja (2005), that presents a model in which optimal board size and composition varies depending on the business of the firm. Based on the model, she concludes that firms in which it easier for outsiders to verify projects tend to have more outsiders on the board than firms with projects that are difficult to verify. Boone et al. (2007) found that a number of variables related to the development of the business of firms that have made an initial public offering have a systematic impact on board size and composition. A more recent paper by Alam et al. (2014) also found that the optimal characteristics of independent board members will differ depending on the business that the firm is operating. They found " . . . that directors tend to reside closer to headquarters when the intangibility of a firm's assets (a proxy for the need to gather soft information) is high." They also found that "more distant" boards base their decisions of CEO dismissals on measured poor performance and they use performance-related compensation to a larger extent than more proximate boards. In cases where the firm's assets are to a larger extent tangible, and the business is thus easier to evaluate for an outsider, application of straightforward performance measures will make the task of the board simpler than in cases where the firm's business is subject to more of genuine uncertainty.

In general, the role of the board in safeguarding unbiasedness of the information disclosed by the firm will differ from one firm to another. If the firm's business is easy to understand and the firm operates in stable markets, there is less scope for management to try to manipulate the market in management's own favor than in cases where the firm is producing complex goods to be sold in markets subject to rapid development. In the latter case, good corporate governance is more important in preventing misuse of managerial talent as well as in guaranteeing disclosure of an accurate view of the firm's value-relevant prospects. A board consisting of knowledgeable persons with untarnished reputation may in that kind of a case have a substantial impact on outside investors' willingness to trade the firm's shares and accordingly on the liquidity of the market for these shares.

## 6. Summary

This paper argues that properly understood good corporate governance will be beneficial for the liquidity of a listed firm's shares. This is largely consistent with empirical evidence from a number of countries around the world. An essential role of the corporate board that shoulders the responsibility for the governance of the firm, is to safeguard that the information dissemination to outside investors is unbiased and timely. An efficient board will, in this fashion, contribute to reducing the adverse selection risk for an outside investor who lacks privileged information and would like to trade the

firm's shares. The interest of a broader group of investors to trade in the firm's shares will thus grow and result in increased liquidity at the stock exchange.

Safeguarding that the information is as unbiased as possible is far from a simple task for the board. There is the obvious issue of calibrating explicit incentives for the top executives of the firm so as not to promote short-sighted attempts to manipulate the share price. Over and above that, proper knowledge of the business and vigilance on behalf of board members is required in combination with a clearly articulated preference for unbiased assessments of the future prospects of the firm's business. For a maximum beneficial impact on the liquidity of the firm's shares in the stock market, this preference for unbiasedness should be the guiding principle in external as well as internal communication of the firm.

**Funding:** This research received no external funding.

**Acknowledgments:** Comments received by two anonymous referees for this journal and participants in the 4th Chapman Conference on Money and Finance, and in the 11th Nordic Corporate Governance Network Conference in Oslo are gratefully acknowledged, as are comments from Clas Wihlborg and from my former PhD student Naufal Alimov.

**Conflicts of Interest:** The author declares no conflict of interest.

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
