# Peer review of "Liquidity and Corporate Governance"

_jrfm, doi:10.3390/jrfm13030054_

Round 1

Reviewer 1 Report

Summary:

This is a discussion paper on the relationship between corporate governance and stock liquidity. The author argues that the positive effect of corporate governance on stock liquidity is the more important for firms which are difficult to understand (i.e., firms with high complexity or uncertainty). Corporate governance is defined using board of directors, and stock liquidity is based on three dimensions i.e., tightness, depth and resilience. I read the paper with interest and find it easy to understand and follow. However, I have some major concerns regarding the contributions of the paper which are as follows;

Suggestions:

My first concern is about the review of literature. Large body of literature exist on the empirical relationship between corporate governance and stock liquidity which has not been discussed and cited in this paper. For example, Foo and Zain (2010) find that board independence and board meetings improve stock liquidity for Malaysian firms. Lei, Lin et al. (2013) show that corporate governance has different effects on liquidity for Chinese firms faced with different types of agency conflicts. Prommin, Jumreornvong et al. (2014) argue that effective governance enhances financial and operational transparency, which in turn, reduces adverse selection. Facing less adverse selection problems, traders provide more liquidity to stocks of well-governed firms in Thailand. Chung, Kim et al. (2012) show that firms with a superior governance structure have greater stock liquidity regardless of the legal origins of the relevant country. Ali, Liu et al. (2017) show that better governed firms have a lower trading cost, smaller price impacts of trade, and higher trading speed. They also show that better governance improves stock liquidity because it is associated with higher information disclosure. In the presence of this large stream of empirical literature on the effect of corporate governance and stock liquidity based on certain firm characteristics (such as type of agency conflict, legal origin of the country, and information disclosure), the paper’s inference “positive effect of corporate governance on stock liquidity is the more important for firms which are difficult to understand” adds little to the understanding of the reader.

The author should also discuss studies on stock liquidity and other aspects of corporate governance such as boardroom gender diversity (Ahmed and Ali 2017), managerial compensation (Feng and Yan 2019), and ownership structure (Prommin, Jumreornvong et al. 2016). Without this, the paper title should be "Board structure and stock liquidity". Yet, the study on boardroom gender diversity is relevant.

Another concern is related to the use of theory. The paper does not state any theory which can be used to understand the link between corporate governance and stock liquidity. In the empirical studies, agency theory has been used to explain the relationship. To strengthen the contribution of the paper, the author should discuss some contrasting or similar theories.

For this kind of discussion paper to have a strong impact on the reader, it would have been much better to have a flow chart or conceptual map linking governance with liquidity based on firm characteristics. I suggest author to think about creating a conceptual figure based on the arguments.

To further improve the contribution of the paper, it would be of interest to the readers how board structure is important to improve each of three dimensions of liquidity (tightness, depth and resiliency) for firms which are difficult to understand (see e.g., Ali, Liu et al. 2017).

I also recommend authors to check stream of literature on the determinants of optimal board structure which have shown that board of directors are endogenously chosen by firms to suit their operating and contracting environment (Raheja 2005, Coles, Daniel et al. 2008, Guest 2008, Linck, Netter et al. 2008). These studies are relevant to the arguments used on page 10 of the paper. If author can clearly differentiate the argument presented in this paper from those already published empirical papers, then I would like to see some empirical analysis showing the positive effect of corporate governance on stock liquidity is stronger (weaker) for complex (simple) firms. 

References

Ahmed, A. and S. Ali (2017). "Boardroom gender diversity and stock liquidity: Evidence from Australia." Journal of Contemporary Accounting & Economics 13(2): 148-165.

Ali, S., et al. (2017). "Corporate governance and stock liquidity dimensions: panel evidence from pure order-driven Australian market." International Review of Economics & Finance 50: 275-304.

Chung, K. H., et al. (2012). "Corporate governance, legal system, and stock market liquidity: Evidence around the world." Asia‐Pacific Journal of Financial Studies 41(6): 686-703.

Coles, J. L., et al. (2008). "Boards: Does one size fit all?" Journal of Financial Economics 87(2): 329-356.

Feng, H. and S. Yan (2019). "CEO incentive compensation and stock liquidity." Review of Quantitative Finance and Accounting 53(4): 1069-1098.

Foo, Y.-B. and M. M. Zain (2010). "Board independence, board diligence and liquidity in Malaysia: A research note." Journal of Contemporary Accounting & Economics 6(2): 92-100.

Guest, P. M. (2008). "The determinants of board size and composition: Evidence from the UK." Journal of Corporate Finance 14(1): 51-72.

Lei, Q., et al. (2013). "Types of agency cost, corporate governance and liquidity." Journal of Accounting and Public Policy 32(3): 147-172.

Linck, J. S., et al. (2008). "The determinants of board structure." Journal of Financial Economics 87(2): 308-328.

Prommin, P., et al. (2014). "The effect of corporate governance on stock liquidity: The case of Thailand." International Review of Economics & Finance 32: 132-142.

Prommin, P., et al. (2016). "Liquidity, ownership concentration, corporate governance, and firm value: Evidence from Thailand." Global Finance Journal 31: 73-87.

Raheja, C. G. (2005). "Determinants of board size and composition: A theory of corporate boards." Journal of Financial and Quantitative Analysis 40(2): 283-306.

Reviewer 2 Report

The paper presents theoretical concepts related to the liquidity and corporate governance. It also provides brief summary of the past empirical studies related to this issue. Unfortunately the literature review is limited and many theoretical concepts are missing. There is no empirical research designed and carried by the Author presented in the paper. Additionally, no new theoretical model or concept is offered by the Author. Thus, the originality and novelty of the paper is limited. It seems that this paper is a preliminary study showing the current state of knowledge on corporate governance and liquidity, that may become a basis for further empirical studies.

Thus, if this papes should stay in this form, it may be recommended to improve the literature review (covering shareholders theory, agency theory and agency costs, asymmetric information and its consequences: adverse selection and moral hazard; effcient market hyphotesis, as well as models and mechanisms of corporate governance). In this way it may become a complete literature review on this issue.

Round 2

Reviewer 1 Report

I appreciate author for the response to my comments and suggestions. Though author is unable to prepare conceptual map (he found it difficult to execute), and argued against the gender diversity (which I think still relevant to be discussed), this version has improved compared to the last version.

Author Response

Thanks again for your valuable comments on my previous version!

Reviewer 2 Report

The Author may add precisely formulated research questions in the introduction to build a coherent structure of the paper.

The literature review was developed, but still it may be expanded, to a form and extent required in this type of conceptual papers.

Chapter liquidity - add discussion on the factors determining stock liquidity in various market conditions; provide references with empirical studies

Chapter corporate governance - board activity and its composition is only one of the mechanisms of corporate governance (other mechanisms may be briefly discussed with regard to various corporate governance models in different countries e.g. Anglo-Saxon model or Continental Europe model); it would be also recommended to add information on the relationship between corporate governance and firm performance;

Author Response

Thank you for your suggestions!

I've now tried to clarify the structure of the paper by rewriting the third paragraph of the introductory section. Since the paper is conceptual it is a bit difficult to formulate specific research questions in the beginning of the paper. 

I agree that the scope of the literature review could be expanded. However, these expansions naturally bring in additional complications and measurement issues that would require careful analysis. 

Liquidity in different market conditions is certainly an interesting topic and I can certainly see  that it could have some impact on the importance of corporate governance for liquidity. However, the issue of proper identification of relevant market conditions is already challenging.

Different governance models internationally is also interesting and I agree that institutional differences in general, most likely, will have an impact on the importance of corporate governance for liquidity, but again it a rather complex field of study due to many other differences between different countries.

Finally the issue of corporate governance and firm performance. There's a huge number of studies in that field, some of them not very well done. In general we would expect performance to improve if good corporate governance succeeds in reducing expected managerial misbehaviour. In addition the market's valuation (not performance related) of the firm should improve if the likelihood of value reducing scandals is perceived to be lower for a well governed firm. 

I hope these comments have clarified why I've chosen to focus on what's now included in the text.